# Child diarrhea in Cambodia: A descriptive analysis of temporal and geospatial trends and logistic regression-based examination of factors associated with diarrhea in children under five years

**Samnang Um** [1]*, **Channnarong Phan**[1], **Sok Sakha**[2], **Leng Dany**[2]

1 National Institute of Public Health (NIPH), Phnom Penh, Cambodia, 2 The Elite Angkor Clinic, Siem Reap, Cambodia

* umsamnang56@gmail.com

**Data Availability Statement:** The Cambodia Demographic and Health Survey (CDHS) was approved by the Cambodia National Ethics

## Abstract

Diarrhea is a global public health problem that is the third leading cause of death in under five years, with an estimated 1.7 billion cases in 2023 and 1.8 million deaths from diarrhea diseases every year. To better understand child diarrhea in Cambodia, we examined to describe temporal and geospatial trends of diarrhea and used multiple logistic regression to analyze its association with individual and household characteristics and diarrhea among children aged 0–59 months using combined data from the Cambodia Demographic and Health Survey for 2005 to 2022. This study included 29,742 weighted children aged 0–59 months; there were 7,220 in 2005, 7,758 in 2010, 7,010 in 2014, and 7,754 in 2022, respectively. The prevalence of diarrhea among children aged 0–59 months decreased from 19.7% in 2005 to 6.2% in 2022. The highest prevalence of childhood diarrhea was observed in Kampong Cham (30.1%), in Mondul Kiri/Ratanak Kiri (24.6%), Battambang/Pailin (20.9%), and Mondul Kiri/Ratanak Kiri (12.9%) for the years 2005, 2010, 2014 and 2022. After adjusting for other variables, factors independently associated with a protective effect against childhood diarrhea included mothers aged 25–34 years (adjusted odds ratio (AOR) = 0.68; 95% CI: 0.48–0.96), 35–49 years (AOR = 0.60; 95% CI: 0.42–0.87), completed higher education (AOR = 0.61; 95% CI: 0.41–0.91), and employed (AOR = 0.79; 95% CI: 0.72–0.96). Children were less likely to have diarrhea if they were older than 36 months, belonged to the richest households (AOR = 0.69; 95% CI: 0.55–0.86), or lived in coastal region (AOR = 0.53; 95% CI: 0.41–0.69). Conversely, children born to mothers who smoke had increased odds of diarrhea (AOR = 1.61; 95% CI: 1.25–2.08), had barrier access to healthcare services (AOR = 1.20; 95% CI: 1.07–1.35), or children aged 6–23 months. Diarrhea remains highly prevalent among children in Cambodia. Public health interventions and policies to alleviate diarrhea should be prioritized to address these factors across geographical.

Committee for Health Research and ICF's Institutional Review Board (IRB) in Rockville, Maryland, USA; the final report of each survey year can be found at the Cambodia National Institute of Statistic website (URL: https://nis.gov.kh/?lang=en) or DHS website at (URL: https://www.dhsprogram.com/publications/index.cfm). The CDHS data are publicly accessible upon request through the DHS website at (URL: https://dhsprogram.com/data/available-datasets.cfm) following permission obtained through an online request outlining the purpose of our investigation and removing all participant personal information. As permission using data provided.

**Funding:** The author(s) received no specific funding for this work.

**Competing interests:** The authors have declared that no competing interests exist.

**Abbreviations:** ACS, American Cancer Society; AOR, adjusted odds ratio; CDHS, Cambodia Demographic Health Survey; EA, enumeration areas; PPS, probability proportional to size.

## Introduction

Diarrhea is defined by the World Health Organization (WHO) as passing loose or watery stool three or more times in 24 hours [1]. Although categorized in several ways, acute and persistent diarrhea are the most common classifications [1]. Persistent diarrhea is defined as diarrhea that starts acutely and lasts for more than 14 days [1]. An infection causes acute diarrhea and typically starts 12 hours to 4 days after exposure and goes away in 3 to 7 days [1]. Diarrhea is usually a symptom of an intestinal infection of various bacterial, viral, and parasitic organisms contained in food, drinking water, or from person to person due to improper hygiene [1]. Diarrhea is a global public health problem that is the third leading cause of death among under five years, with an estimated 1.7 billion childhood diarrhea cases in 2023 [1]. An estimated 1.8 million children under the age of five die from diarrheal illnesses every year, with over 80% of these deaths suffering in developing nations [3, 4], including Southeast Asia [3, 4]. In Southeast Asia, diarrheal diseases continue to rank among the top five causes of death, contributing to 6% of all deaths [2]. Even though still morbidity and mortality of children due to diarrhea are high in Cambodia, the percentage of children under five who had experienced diarrhea two weeks before the survey was 19.5% in 2005; this number slightly decreased to 15% in 2010 and 13% in 2014, and it also dramatically reduced to 6% in 2022 [3–6].

Additionally, the under-five mortality rate (U5M) in Cambodia has decreased from 124 deaths per 1,000 live births in 2000 to 16 deaths per 1,000 live births in 2022 [3–6]. However, its mortality is still unacceptably high due to various causes that are preventable, with diarrhea accounting for up to 8% of all deaths among children under the age of five [7]. The under-5 mortality rates are higher in rural areas (15 deaths per 1,000 live births and 20 deaths per 1,000 live births, respectively) than in urban areas. Under-five mortality rates were highest in Ratanak Kiri province (43 deaths per 1,000 live births), Mondul Kiri province (43 deaths per 1,000 live births), Preah Vihear, and Kompong Chhang (each 36 deaths per 1,000 live births) [3–6]. Also, U5M is still higher than in the neighboring countries such as Vietnam and Thailand [8]. Diarrheal illness is a significant public health concern in Cambodia. It is still one of the top 10 causes of disability-adjusted life years (DALYs) across all age groups and a substantial cause of death for children [2]. Many associated factors with diarrhea included the age of children, place of residence, maternal education, and household economic condition [9–11]. Children living in households using improved drinking water, improved toilet facilities, and safe waste disposal were associated with lower odds of diarrhea [9–11].

Furthermore, breastfeeding practices, eating habits, and good handwashing practices were found to be significantly associated with diarrhea in children [9–11]. The incidence of diarrhea has been reported as compared to other developing countries, including malnutrition and early childhood mortality, which are unacceptably high in Cambodia. Furthermore, to prevent and control diarrhea in the future, it is necessary to know the trends of diarrhea over time, the geographical distributions, and associated risk factors. This enhanced understanding will help to identify sub-populations at greater risk for diarrhea and prioritize geographic areas for targeted public health interventions. To our authors' knowledge, no published peer-reviewed studies have been conducted nationally to identify determinants of diarrhea in Cambodia over time. Therefore, this study aims to explore descriptive temporal and geographical trends of diarrhea among under-five children across 2005, 2010, 2014, and 2022 CDHS surveys, identify factors associated with diarrhea among under-five children, and identify the factors that contributed positively or negatively associated with diarrhea among under-five children. Understanding these factors further supports policy development and programs with more effective strategies and interventions in Cambodia to reduce the prevalence of diarrhea among under-five children and associated health risks such as childhood mortality.

## Methods

### Data source

We used children's data from the 2005, 2010, 2014, and 2022 CDHS. The CDHS is a nationally representative population-based household survey conducted roughly every five years. The survey typically uses two-stage stratified cluster sampling to collect the samples from all provinces divided into urban and rural areas. In the first stage, clusters, or enumeration areas (EAs) representing the entire country, are randomly selected from the sampling frame using probability proportional to cluster size (PPS). The second stage then systematically samples households listed in each cluster or EA. Then, interviews were conducted with women aged 15–49 who gave birth five years before the survey in the selected households. Data about the maternal and children demographic, household characteristics, health status, behaviors, and access to healthcare were collected when the enumerators identified children in the selected households. Details of the CDHS survey reported have been described elsewhere [3–6]. We limited our analysis to live births in the last five years before the surveys and children collected on diarrhea. This resulted in a total sample size of 30,314 (**29,742 weighted**) children 0–59 months, including in the final analysis (see **Fig 1**).

### Measurements

**Outcome variable.** The outcome variable of this study was children under five years old having diarrhea for the last two weeks before the data collection, which was coded as "Yes = 1" and "No = 0".

**Indenpedent variables. Maternal characteristics** included age (coded as 1 = 15–18 (reference), 2 = 19–24, 3 = 25–34, and 4 = 35–49 years); education (coded as 0 = no education (reference), 1 = primary, and 2 = secondary, and 3 = higher); employment (coded as 0 = not working (reference), 1 = working); smoking (coded as 0 = non-smoker and 1 = smoker); media exposure (coded as 0 = no and 1 = yes); and healthcare barriers was coded as a dichotomous variable with No Barriers = 0 and 1 or more barriers = 1 (possible barriers reported included distance, money, and waiting time).

**Child's characteristics** included sex of child (coded as 0 = girl and 1 = boy); child age (coded as 0 = 0–5 (reference), 1 = 6–11, 2 = 12–23, 3 = 24–35, 4 = 36–47, and 5 = 48–60 months); birth order (coded as 1 = first child (reference), 2 = 2–3, 3 = 4–5 and 4 = 6 or more); twin birth (coded as 0 = single and 1 = twin); still breastfeeding (coded as 0 = no and 1 = yes); and received measles vaccinated (coded as 0 = no and 1 = yes); and vitamin A supplementation within the last six months (coded as 0 = no and 1 = yes).

**Household characteristics** included the DHS wealth index, coded as an ordinal level variable with richest = 1 (reference), richer = 2, middle = 3, poorer = 4, and poorest = 5 (we opted to use the wealth index that was provided by each respective survey year of the CDHS as opposed to calculating a wealth index across the pooled data. This decision was based on prior research that found wealth status designation was comparable across the CDHS-provided indices and an index based on data pooled across years) [3–6]; source of drinking water (coded as 1 = unimproved and 0 = improved), type of toilet (coded as 1 = unimproved and 0 = improved; and residence (coded as 1 = urban and 2 = rural). Cambodia's domains/provinces were regrouped for analytic purposes into a categorical variable with five geographical regions that were coded as Phnom Penh capital city = 1 (reference), Plains = 2, Tonle Sap = 3, costal/sea = 4, and mountains = 5 (Phnom Penh capital city; Plains included Kampong Cham/Tbong Khmum, Kandal, Prey Veng, Svay Rieng, and Takeo; Tonle Sap included Banteay Meanchey, Kampong Chhnang, Kampong Thom, Pursat, Siem Reap,

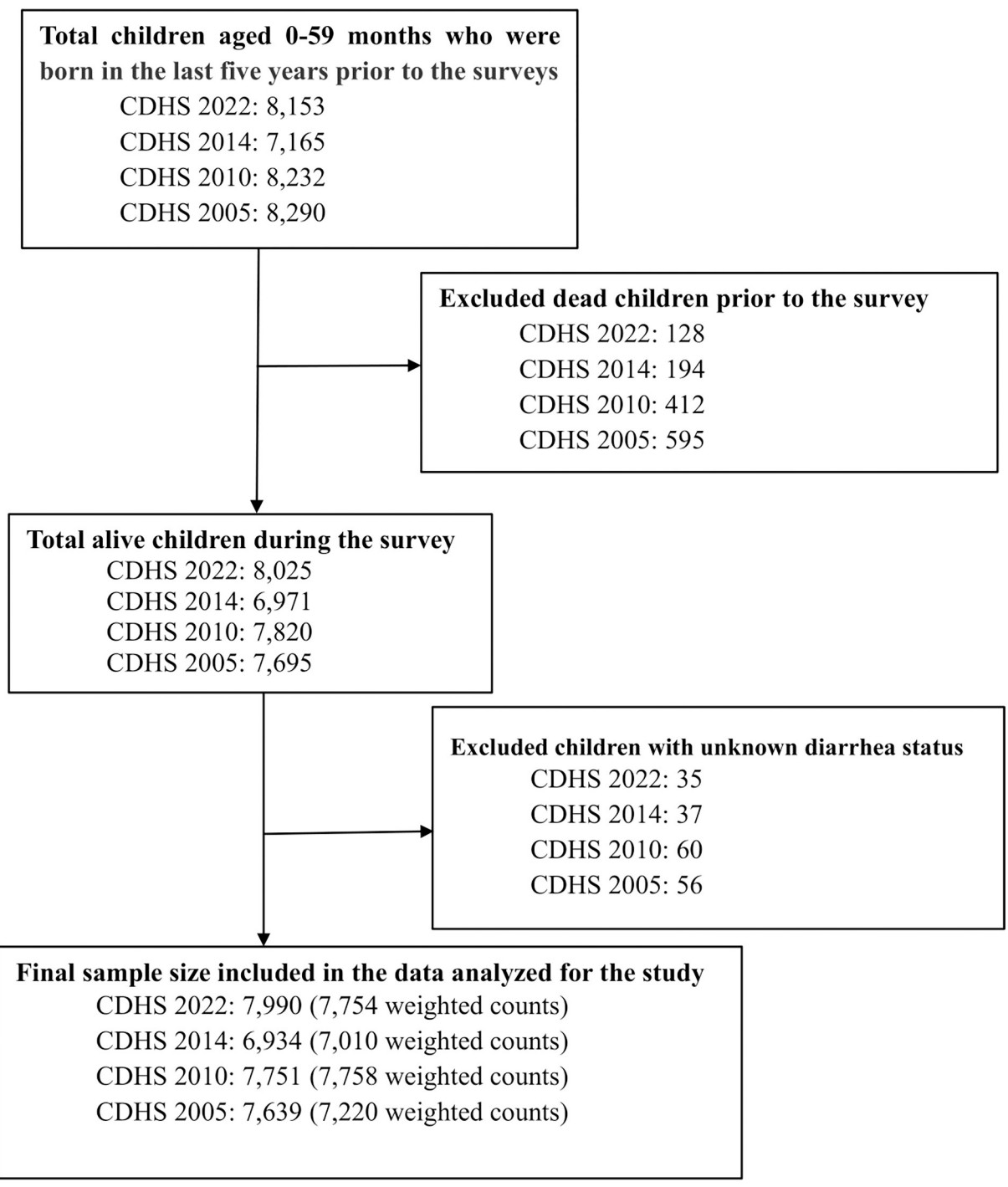

**Fig 1. The total sample size from the 2005, 2010, 2014, and 2022 CDHS.**

Battambang, Pailin, and Otdar Meanchey; Coastal/sea included Kampot, Kep, Preah Siha-nouk, and Koh Kong; and Mountains included Kampong Speu, Kratie, Preah Vihear, Stung Treng, Mondul Kiri, and Ratanak Kiri). Survey year was coded as 1 = 2005 (reference), 2 = 2010, 3 = 2014 and 4 = 2022.

## Statistical analysis

Statistical analyses were performed using STATA version 18 (Stata Crop 2023, College Station, TX) [12]. We formally incorporated the DHS's complex sample design using the "survey" package; all estimations were carried out using the svy command in our descriptive and logistic regression analyses. Key maternal, child, and household characteristics were described using weighted frequency distributions for specific survey years and pooled together for all years. A temporal series of maps illustrating provincial variations in the prevalence of diarrhea over time was created using ArcGIS software version 10.3 [13]. The Cambodian shapefile of provinces was obtained from the United Nations for Coordination of Humanitarian Affairs (OCHA) at (URL: https://data.humdata.org/dataset/cod-ab-khm). Data quality assurance and minor data cleaning followed standard procedures in preparation for statistical analysis [14].

Bivariate analysis using the Chi-square test assessed associations between the variables of interest (including maternal-child, household characteristics, and geographical regions) and diarrhea status. The multiple logistic regression analysis used variables associated with diarrhea status with a p-value $\leq 0.05$ [15, 16]. Women's age, place of residence, and survey years were included in the multiple regression based on the literature and prior knowledge [9]. Simple logistic regression was used to analyze associations between diarrhea and maternal, children, household characteristics, and geographical regions. Results are reported as unadjusted odds ratios (COR) with 95% confidence intervals (CI) and corresponding p-values. Multiple logistic regression analysis was then used to assess independent factors associated with diarrhea after adjusting for other covariate variables in the final model. The final multiple logistic regression results were reported as adjusted odds ratios (AOR) with 95% confidence intervals and corresponding p-values. The final multiple logistic regression model results were considered statistically significant based on a p-value less than 0.05 and 95% confidence intervals that did not cross an AOR of 1.0. Potential multi-collinearity between right-hand-sided variables was checked using a continuous version of the outcome variable and evaluating VIF scores after fitting a regression model (**S1 Table**).

## Ethical consideration

The Cambodia Demographic and Health Survey (CDHS) 2005–2022 was approved by the Cambodia National Ethics Committee for Health Research and ICF's Institutional Review Board (IRB) in Rockville, Maryland, USA. The study participants were informed about the survey objectives before their interview, and written informed consent was obtained from their parents or guardians before data collection for each participant under 18 years old. CDHS data are publicly accessible upon request through the DHS website at (URL: https://dhsprogram.com/data/available-datasets.cfm) following permission obtained through an online request outlining the purpose of our investigation and removing all participant personal information. The details about the CDHS's report [3–6].

## Results

The combined data included 29,742 children aged 0–59 months; there were 7,220 in 2005, 7,758 in 2010, 7,010 in 2014, and 7,754 in 2022, respectively. Overall, 21.7% had a mother above 35 years old, 31.3% had a mother who had at least completed secondary education, 16.6% did not receive formal schooling, 37.2% had an unemployed mother, and 2.9% had a mother who smoked. 34.1% of children were first-order births, and 6.7% were sixth- or high-order births. 1.4% of children were twins, 49.5% were girls, 46.3% were still breastfeeding, 60.9% received the measles vaccine, and 49.1% were Vitamin A intake. Regarding households characterized by children, 78.8% resided in rural areas, 24.6% belonged to the poorest families,

45.8% belonged to households drinking unimproved water, and 51.0% were unimproved toilets (see **Table 1**).

The overall prevalence trend of children aged 0–59 months having diarrhea in the previous two weeks significantly declined from 19.7% in 2005 to 15.0% in 2010 to 12.9%% in 2014 and decreased to 6.2% in 2022 (see **Table 1**). When stratified by age, the diarrhea trend peaked at around 30% among children aged 6–23 months in 2005. However, this trend among this age group gradually declined to less than 24% in 2010, 17% in 2014, and 12% in 2022 (see **Fig 2**).

The extent of this downward trend varied across the five geographical regions; the highest prevalence of diarrhea was in the Plain region in 2005 (21.7%). The Phnom Penh capital had the lowest prevalence of diarrhea across all survey years, with prevalence levels in 2022 that were lower than those of other regions (see **Table 2**). Diarrhea was observed highest among children in Kampong Cham (30.1%), in Mondul Kiri/Ratanak Kiri (24.6%), in Battambang/Pailin (20.9%), and Mondul Kiri/Ratanak Kiri (12.9%) for 2005, 2010, 2014 and 2022, respectively (see **Fig 3**). The lowest prevalence of diarrhea among children was observed in Preah Sihanouk and Koh Kong provinces (9.1%), in Kampong Speu (7.5%), in Prey Veng (4.7%), and in Otdar Meanchey (1%) for 2005, 2010, 2014 and 2022.

Preliminary analyses included the evaluation of cross-tabular frequency distributions combined with Chi-square tests (see **Table 3**). Maternal age was significantly associated with the risk of diarrhea in children in all survey years; the most significant percentage of children with diarrhea were those with younger mothers aged less than 19 years (25.8% in 2005, 27.8% in 2010, 20.7% in 2014 and 13.3%). Maternal educational attainment was significantly associated with diarrhea in children in all survey years; the most significant percentage of children with diarrhea were generally those whose mothers had no formal education (21.5% in 2005, 17.5% in 2010, 13.4% in 2014, and 6.3% in 2022). Maternal unemployment was also significantly associated with diarrhea in children in survey years 2005, 2014, and 2022; the prevalence of diarrhea among children who had unemployed mothers was 21.5% in 2005, 15.2% in 2014, and 7.6% in 2022. Maternal smoking of cigarettes was also significantly associated with diarrhea in children in survey years 2010 and 2014; the prevalence of diarrhea among children who had mothers smoking cigarettes was 25.0% in 2010 and 23.9% in 2014. Maternal who had at least one barrier access to healthcare were also significantly associated with diarrhea in children in all surveys; the most significant percentage of children with diarrhea were generally those whose mothers had at least one barrier access to healthcare (20.5% in 2005, 15.8% in 2010, 13.4% in 2014, and 7.0% in 2022). Diarrhea was related to children's ages, with a higher prevalence in the age group of 6 to 11 months (6–11 months: 32% in 2005, 26.5% in 2010, 20% in 2014 and 11.6% in 2022; 12–23 months: 28.0%, 21.1%, 19.0% and 9.4%, respectively). In addition, diarrhea among boys was higher than among girls: 22.0% in 2005 and 16.1% in 2014. Breastfeeding children had a higher prevalence of diarrhea (21.8% in 2005, 17.7% in 2010, 14.2% in 2014, and 8.7% in 2022). Measles vaccination was also associated with lower diarrhea in children at 18.5% in 2005, 18.6% in 2010, and 14.2% in 2014. The proportion of children with diarrhea in rural areas was higher than in urban areas: 20.2% in 2005, 15.8% in 2010, and 7.2% in 2022. The proportion of children with diarrhea was lowest among children in the Coastal region in the survey years 2005, 2010, and 2014: 10.4% in 2005, 14.9% in 2010, 9.7% in 2014, and Phnom Penh capital city was 2.5% in 2022, respectively. Diarrhea was also associated with household toilet facilities in survey years 2005 and 2010; 21.1% in 2005 and 16.6% in 2010 of children in households that reported having non-improved toilet facilities were diarrhea. The proportion of children with diarrhea was highest among children in the poorest households across all the survey years: 22.6% in 2005, 18.5% in 2010, 16.1% in 2014, and 8.2% in 2022, respectively.

**Table 1. Descriptive statistics of individual and household socio-demographic characteristics.**

| Variables | | 2005 (n = 7,220) | | 2010 (n = 7,758) | | 2014 (n = 7,010) | | 2022 (n = 7,754) | | 2005–2022 (n = 29,742) | |
|---|---|---|---|---|---|---|---|---|---|---|---|
| | | Freq. | % | Freq. | % | Freq. | % | Freq. | % | Freq. | % |
| **Maternal Characteristic** | | | | | | | | | | | |
| Age (years) | | | | | | | | | | | |
| | <19 | 89 | 1.2 | 90 | 1.2 | 92 | 1.2 | 98 | 1.3 | 370 | 1.2 |
| | 19–24 | 1,918 | 26.6 | 1,825 | 23.5 | 1,828 | 23.6 | 1,457 | 18.8 | 7,029 | 23.6 |
| | 25–34 | 3,321 | 46.0 | 4,282 | 55.2 | 4,022 | 51.8 | 4,269 | 55.0 | 15,894 | 53.4 |
| | 35–49 | 1,891 | 26.2 | 1,561 | 20.1 | 1,068 | 13.8 | 1,930 | 24.9 | 6,450 | 21.7 |
| Educational | | | | | | | | | | | |
| | No education | 1,720 | 23.8 | 1,419 | 18.3 | 962 | 12.4 | 841 | 10.8 | 4,942 | 16.6 |
| | Primary | 4,251 | 58.9 | 4,385 | 56.5 | 3,675 | 47.4 | 3,183 | 41.0 | 15,494 | 52.1 |
| | Secondary | 1,207 | 16.7 | 1,833 | 23.6 | 2,175 | 28.0 | 3,160 | 40.7 | 8,375 | 28.2 |
| | Higher | 42 | 0.6 | 120 | 1.5 | 198 | 2.6 | 570 | 7.3 | 931 | 3.1 |
| Employment | | | | | | | | | | | |
| | Not working | 2,952 | 40.9 | 2,629 | 33.9 | 2,523 | 36.0 | 2,958 | 38.1 | 11,061 | 37.2 |
| | Working | 4,267 | 59.1 | 5,130 | 66.1 | 4,483 | 64.0 | 4,796 | 61.9 | 18,676 | 62.8 |
| Smoking | | | | | | | | | | | |
| | No-smoker | 6,878 | 95.3 | 7,546 | 97.3 | 6,813 | 97.2 | 7,653 | 98.7 | 28,890 | 97.1 |
| | Smoker | 342 | 4.7 | 212 | 2.7 | 197 | 2.8 | 100 | 1.3 | 852 | 2.9 |
| Media exposure | | | | | | | | | | | |
| | No | 6,846 | 94.8 | 7,427 | 95.7 | 6,808 | 97.1 | 7,654 | 98.7 | 28,736 | 96.6 |
| | Yes | 374 | 5.2 | 331 | 4.3 | 202 | 2.9 | 99 | 1.3 | 1,006 | 3.4 |
| Healthcare Barriers | | | | | | | | | | | |
| | No barrier | 1,084 | 15.0 | 2,181 | 28.1 | 1,753 | 25.0 | 3,160 | 40.8 | 8,178 | 27.5 |
| | One or More Barriers | 6,136 | 85.0 | 5,578 | 71.9 | 5,257 | 75.0 | 4,594 | 59.2 | 21,564 | 72.5 |
| **Child Characteristics** | | | | | | | | | | | |
| Had diarrhea in the past two weeks | | | | | | | | | | | |
| | Not diarrhea | 5,800 | 80.3 | 6,597 | 85.0 | 6,108 | 87.1 | 7,276 | 93.8 | 25,782 | 86.7 |
| | Diarrhea | 1,420 | 19.7 | 1,161 | 15.0 | 902 | 12.9 | 477 | 6.2 | 3,960 | 13.3 |
| Age(months) | | | | | | | | | | | |
| | 0-5m | 743 | 10.3 | 710 | 9.2 | 732 | 10.4 | 760 | 9.8 | 2,946 | 9.9 |
| | 6-11m | 769 | 10.7 | 825 | 10.6 | 759 | 10.8 | 809 | 10.4 | 3,163 | 10.6 |
| | 12-23m | 1,513 | 21.0 | 1,608 | 20.7 | 1,457 | 20.8 | 1,642 | 21.2 | 6,220 | 20.9 |
| | 24-35m | 1,411 | 19.5 | 1,601 | 20.6 | 1,360 | 19.4 | 1,496 | 19.3 | 5,866 | 19.7 |
| | 36-47m | 1,409 | 19.5 | 1,517 | 19.6 | 1,332 | 19.0 | 1,519 | 19.6 | 5,777 | 19.4 |
| | 48-60m | 1,374 | 19.0 | 1,498 | 19.3 | 1,370 | 19.5 | 1,528 | 19.7 | 5,770 | 19.4 |
| Sex | | | | | | | | | | | |
| | Boys | 3,584 | 49.6 | 4,006 | 51.6 | 3,506 | 50.0 | 3,935 | 50.7 | 15,031 | 50.5 |
| | Girls | 3,636 | 50.4 | 3,752 | 48.4 | 3,504 | 50.0 | 3,819 | 49.3 | 14,711 | 49.5 |
| Birth order | | | | | | | | | | | |
| | 1 child | 2,008 | 27.8 | 2,664 | 34.3 | 2,734 | 39.0 | 2,747 | 35.4 | 10,152 | 34.1 |
| | 2–3 child | 2,866 | 39.7 | 3,365 | 43.4 | 3,185 | 45.4 | 4,172 | 53.8 | 13,588 | 45.7 |
| | 4–5 child | 1,389 | 19.2 | 1,117 | 14.4 | 801 | 11.4 | 699 | 9.0 | 4,006 | 13.5 |
| | ≥ 6 child | 958 | 13.3 | 613 | 7.9 | 290 | 4.1 | 135 | 1.7 | 1,996 | 6.7 |
| Twin | | | | | | | | | | | |
| | Single | 7,129 | 98.7 | 7,649 | 98.6 | 6,889 | 98.3 | 7,672 | 98.9 | 29,339 | 98.6 |
| | Multiple | 91 | 1.3 | 109 | 1.4 | 121 | 1.7 | 82 | 1.1 | 403 | 1.4 |

*(Continued)*

**Table 1.** (Continued)

| Variables | | 2005 | | 2010 | | 2014 | | 2022 | | 2005–2022 | |
|---|---|---|---|---|---|---|---|---|---|---|---|
| | | (n = 7,220) | | (n = 7,758) | | (n = 7,010) | | (n = 7,754) | | (n = 29,742) | |
| Still breastfeeding | | | | | | | | | | | |
| | No | 3,153 | 43.7 | 3,898 | 50.2 | 3,836 | 54.7 | 5,091 | 65.7 | 15,978 | 53.7 |
| | Yes | 4,067 | 56.3 | 3,860 | 49.8 | 3,174 | 45.3 | 2,663 | 34.3 | 13,763 | 46.3 |
| Measles' vaccine | | | | | | | | | | | |
| | Not vaccinated | 2,621 | 36.3 | 2,349 | 30.3 | 1,999 | 28.5 | 4,660 | 60.1 | 11,629 | 39.1 |
| | Vaccinated | 4,599 | 63.7 | 5,409 | 69.7 | 5,011 | 71.5 | 3,094 | 39.9 | 18,113 | 60.9 |
| Vitamin A supplement | | | | | | | | | | | |
| | No | 4,836 | 67.0 | 2,771 | 35.7 | 2,560 | 36.5 | 4,960 | 64.0 | 15,127 | 50.9 |
| | Yes | 2,383 | 33.0 | 4,987 | 64.3 | 4,451 | 63.5 | 2,794 | 36.0 | 14,615 | 49.1 |
| **Household Characteristics** | | | | | | | | | | | |
| Wealth index | | | | | | | | | | | |
| | Poorest | 1,930 | 26.7 | 2,035 | 26.2 | 1,684 | 24.0 | 1,668 | 21.5 | 7,317 | 24.6 |
| | Poorer | 1,639 | 22.7 | 1,657 | 21.4 | 1,401 | 20.0 | 1,445 | 18.6 | 6,141 | 20.6 |
| | Middle | 1,273 | 17.6 | 1,417 | 18.3 | 1,331 | 19.0 | 1,439 | 18.6 | 5,460 | 18.4 |
| | Richer | 1,171 | 16.2 | 1,355 | 17.5 | 1,210 | 17.3 | 1,654 | 21.3 | 5,390 | 18.1 |
| | Richest | 1,206 | 16.7 | 1,295 | 16.7 | 1,384 | 19.7 | 1,549 | 20.0 | 5,434 | 18.3 |
| Place of residence | | | | | | | | | | | |
| | Urban | 1,017 | 14.1 | 1,232 | 15.9 | 1,014 | 14.5 | 3,048 | 39.3 | 6,310 | 21.2 |
| | Rural | 6,203 | 85.9 | 6,527 | 84.1 | 5,996 | 85.5 | 4,706 | 60.7 | 23,432 | 78.8 |
| Region | | | | | | | | | | | |
| | Phnom Penh | 580 | 8.0 | 622 | 8.0 | 603 | 8.6 | 1,131 | 14.6 | 2,936 | 9.9 |
| | Plain | 2,766 | 38.3 | 3,068 | 39.5 | 2,580 | 36.8 | 2,576 | 33.2 | 10,990 | 37.0 |
| | Tonle Sap | 2,381 | 33.0 | 2,509 | 32.3 | 2,251 | 32.1 | 2,504 | 32.3 | 9,644 | 32.4 |
| | Coastal | 548 | 7.6 | 503 | 6.5 | 433 | 6.2 | 458 | 5.9 | 1,942 | 6.5 |
| | Mountain | 945 | 13.1 | 1,056 | 13.6 | 1,144 | 16.3 | 1,085 | 14.0 | 4,229 | 14.2 |
| Drinking water | | | | | | | | | | | |
| | Improved | 3,755 | 52.0 | 4,222 | 54.4 | 3,516 | 50.2 | 4,629 | 59.7 | 16,122 | 54.2 |
| | Unimproved | 3,464 | 48.0 | 3,536 | 45.6 | 3,494 | 49.8 | 3,125 | 40.3 | 13,619 | 45.8 |
| Toilet facility | | | | | | | | | | | |
| | Improved | 1,567 | 21.7 | 2,772 | 35.7 | 3,488 | 49.8 | 6,748 | 87.0 | 14,574 | 49.0 |
| | Non-Improved | 5,653 | 78.3 | 4,986 | 64.3 | 3,522 | 50.2 | 1,006 | 13.0 | 15,167 | 51.0 |

Notes: Survey weights were applied to obtain weighted percentages across survey years and combined data. *Phnom Penh capital city; Plains: Kampong Cham, Tbong Khmum, Kandal, Prey Veng, Svay Rieng, and Takeo; Tonle Sap: Banteay Meanchey, Kampong Chhnang, Kampong Thom, Pursat, Siem Reap, Battambang, Pailin, and Otdar Meanchey; Coastal/sea: Kampot, Kep, Preah Sihanouk, and Koh Kong; Mountains: Kampong Speu, Kratie, Preah Vihear, Stung Treng, Mondul Kiri, and Ratanak Kiri.

As shown in **Table 4**, several factors were independently associated with children's decreased odds of having diarrhea in the last two weeks. These factors included children born to mothers aged 25–34 years (AOR = 0.68; 95% CI: 0.48–0.96), 35–49 years (AOR = 0.60; 95% CI: 0.42–0.87), born to mothers with secondary education (AOR = 0.84; 95% CI: 0.72–0.98), higher educated (AOR = 0.61; 95% CI: 0.41–0.91), employed mothers (AOR = 0.79; 95% CI: 0.72–0.96). For age, children aged 36–47 months (AOR = 0.64; 95% CI: 0.52–0.79) and 48–60 months (AOR = 0.52; 95% CI: 0.42–0.66) were less likely to be diarrhea than those aged 0–5 months. Also, children had lower odds of having diarrhea if they were from middle households (AOR = 0.77; 95% CI: 0.67–0.89), richer households (AOR = 0.79; 95% CI: 0.66–0.93), richest

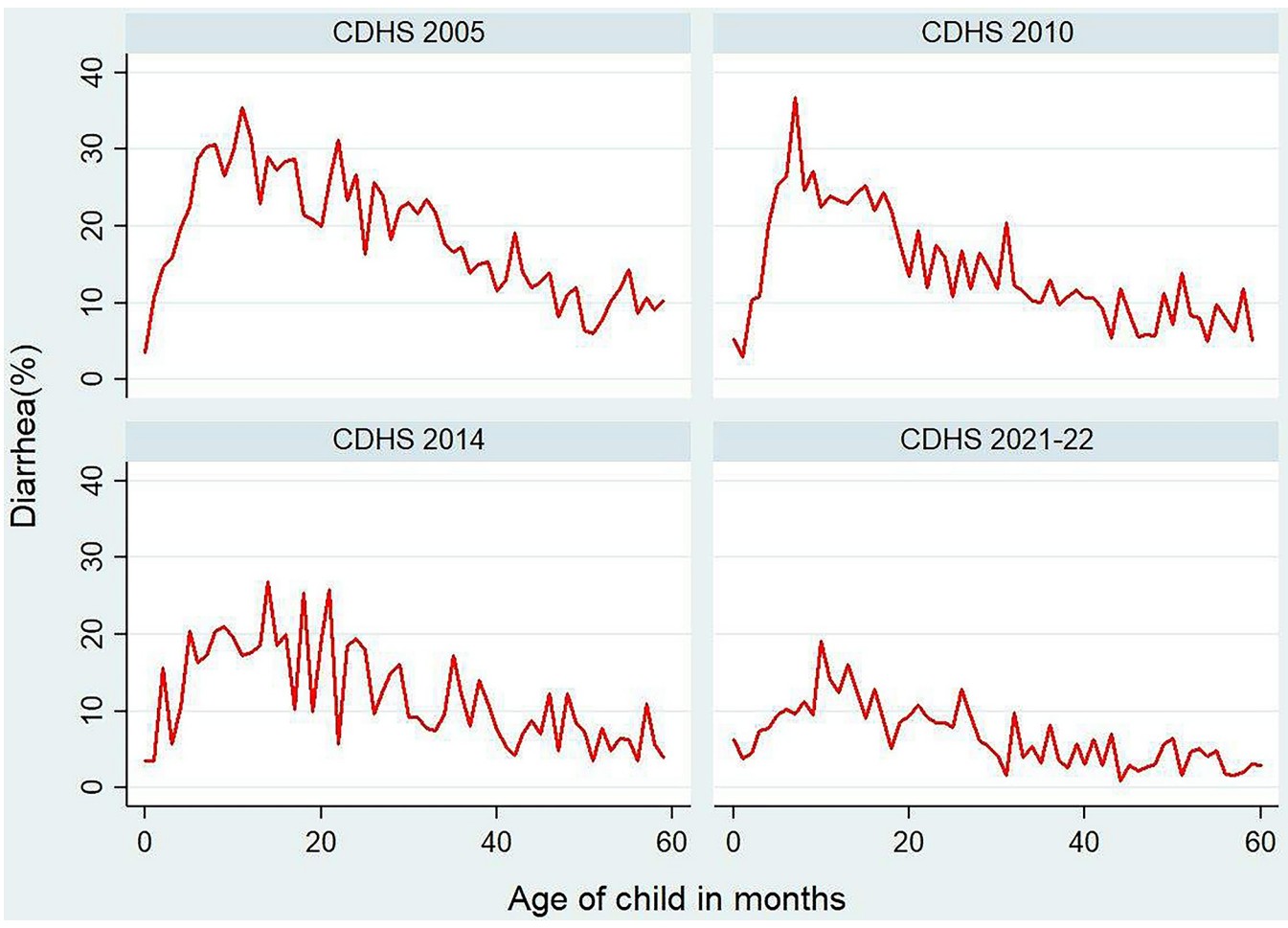

**Fig 2. Age distribution trends of diarrhea among children 0–59 months in the last two weeks by survey years, CDHS 2005 to 2022.** Notes: Survey weights applied to obtain weighted percentages across survey years and combined data.

households (AOR = 0.69; 95% CI: 0.55–0.86). In addition, children had lower odds of having diarrhea if they were from the coastal region (AOR = 0.53; 95% CI: 0.41–0.69), mountain region (AOR = 0.67; 95% CI: 0.53–0.85), survey years in 2010 (AOR = 0.74; 95% CI: 0.64–0.85), 2014 (AOR = 0.62; 95% CI: 0.54–0.72), 2022 (AOR = 0.32; 95% CI: 0.26–0.38). Converly, children were more likely to have diarrhea if they were born to mothers who smoked (AOR = 1.61; 95% CI: 1.25–2.08) compared to those born to non-smokers. Additionally,

**Table 2. Overall and geographical regional trends of diarrhea among children 0–59 months in the last two weeks by survey years, CDHS 2005 to 2022.**

| Region | | 2005 | 2010 | 2014 | 2021–22 |
|---|---|---|---|---|---|
| | Phnom Penh | 19.0 | 11.9 | 17.7 | 2.5 |
| | Plain | 21.7 | 15.6 | 10.8 | 6.8 |
| | Tonle Sap | 19.1 | 16.6 | 13.7 | 6.9 |
| | Coastal | 10.4 | 14.9 | 9.7 | 5.2 |
| | Mountain | 20.7 | 11.0 | 14.7 | 7.2 |

Notes: Survey weights applied to obtain weighted percentages across survey years and combined data.

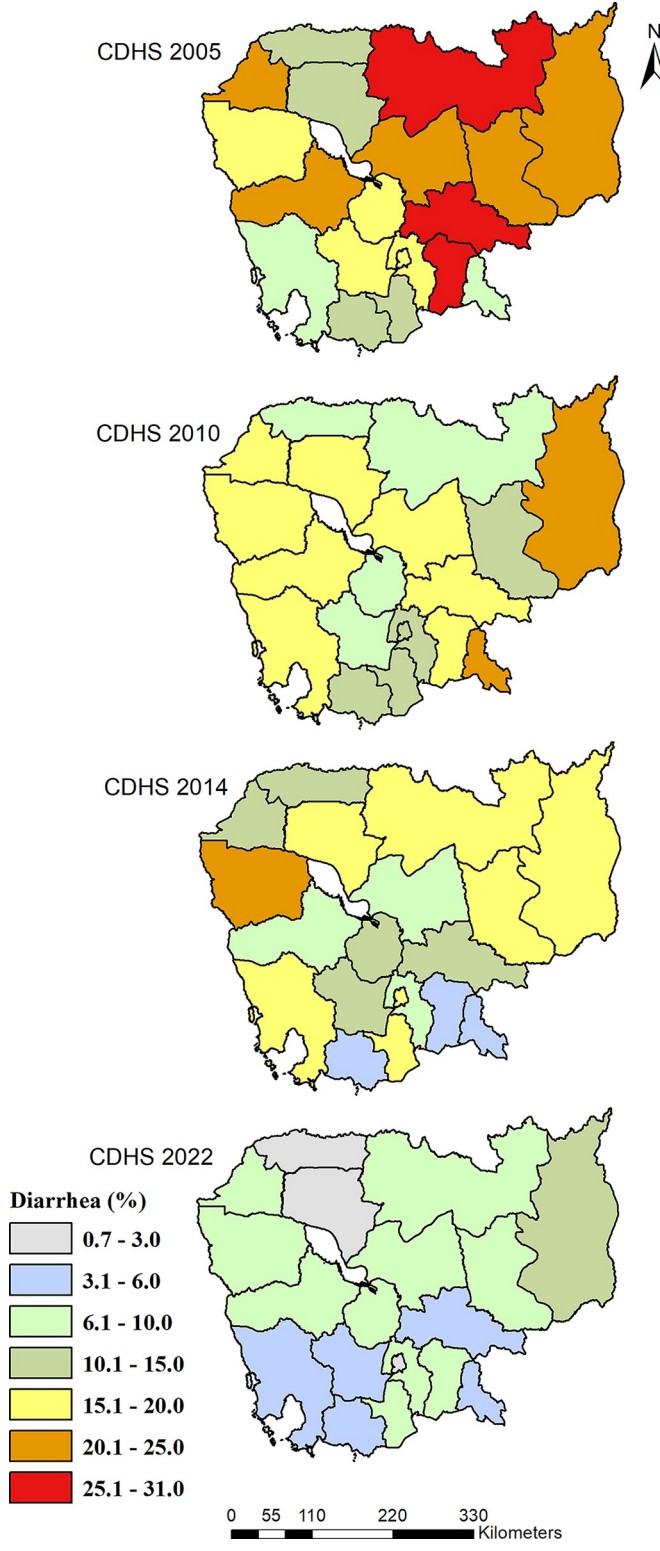

**Fig 3. Geographical distribution of diarrhea among children 0–59 months in the last two weeks by survey years, CDHS 2005 to 2022.** Notes: Survey weights were applied to obtain weighted percentages across survey years and combined data. The map was created using ArcGIS software version 10.3 [13]. The Cambodian shapefile of provinces was obtained from the United Nations for Coordination of Humanitarian Affairs (OCHA) at (URL: https://data. humdata.org/dataset/cod-ab-khm).

**Table 3. Chi-square test results for associations between child diarrhea and child, maternal, and household characteristics.**

| Variables | | 2005 (n = 7,220) | | 2010 (n = 7,758) | | 2014 (n = 7,010) | | 2022 (n = 7,754) | |
|---|---|---|---|---|---|---|---|---|---|
| | | % Diarrhea | P value | % Diarrhea | P value | % Diarrhea | P value | Diarrhea | P value |
| **Maternal Characteristic** | | | | | | | | | |
| **Age (years)** | | | | | | | | | |
| | <19 | 25.8 | 0.029 | 27.8 | <0.001 | 20.7 | <0.001 | 13.3 | <0.001 |
| | 19–24 | 22.3 | | 18.8 | | 16.3 | | 9.0 | |
| | 25–34 | 19.2 | | 14.0 | | 12.2 | <0.001 | 5.1 | |
| | 35–49 | 17.5 | | 12.4 | | 8.9 | | 6.0 | |
| **Educational** | | | | | | | | | |
| | No education | 21.5 | 0.008 | 17.5 | <0.001 | 13.4 | | 6.3 | 0.001 |
| | Primary | 20.1 | | 15.8 | | 13.0 | | 7.6 | |
| | Secondary | 15.9 | | 11.6 | | 12.7 | | 5.2 | |
| | Higher | 9.5 | | 6.7 | | 9.6 | | 3.3 | |
| **Employment** | | | | | | | | | |
| | Not working | 21.9 | 0.002 | 15.6 | 0.357 | 15.2 | 0.004 | 7.6 | <0.001 |
| | Working | 18.1 | | 14.6 | | 11.6 | | 5.3 | |
| **Smoking** | | | | | | | | | |
| | No-smoker | 19.5 | 0.145 | 14.7 | 0.026 | 12.5 | <0.001 | 6.1 | 0.147 |
| | Smoker | 22.8 | | 25.0 | | 23.9 | | 9.0 | |
| **Media exposure** | | | | | | | | | |
| | No | 19.9 | 0.103 | 15.1 | 0.096 | 13.0 | 0.162 | 6.2 | 0.375 |
| | Yes | 15.0 | | 11.2 | | 8.9 | | 4.0 | |
| **Healthcare Barriers** | | | | | | | | | |
| | No barrier | 14.9 | <0.001 | 12.9 | 0.02 | 11.2 | 0.038 | 4.9 | 0.003 |
| | One or More Barriers | 20.5 | | 15.8 | | 13.4 | | 7.0 | |
| **Child Characteristics** | | | | | | | | | |
| **Age(months)** | | | | | | | | | |
| | 0-5m | 17.6 | <0.001 | 14.2 | <0.001 | 12.8 | <0.001 | 5.9 | <0.001 |
| | 6-11m | 32.0 | | 26.5 | | 20.0 | | 11.6 | |
| | 12-23m | 28.0 | | 21.1 | | 19.0 | | 9.4 | |
| | 24-35m | 20.6 | | 13.8 | | 13.8 | | 5.3 | |
| | 36-47m | 13.8 | | 9.8 | | 7.5 | | 3.8 | |
| | 48-60m | 9.9 | | 8.8 | | 6.6 | | 3.1 | |
| **Sex** | | | | | | | | | |
| | Boys | 22.0 | <0.001 | 16.1 | 0.029 | 13.5 | 0.22 | 6.4 | 0.532 |
| | Girls | 17.4 | | 13.8 | | 12.2 | | 5.9 | |
| **Birth order** | | | | | | | | | |
| | 1 child | 19.5 | 0.546 | 15.5 | 0.88 | 14.1 | 0.135 | 6.5 | 0.321 |
| | 2–3 child | 20.1 | | 14.8 | | 12.4 | | 5.7 | |
| | 4–5 child | 18.2 | | 14.6 | | 12.0 | | 7.3 | |
| | ≥ 6 child | 20.8 | | 14.4 | | 8.6 | | 8.1 | |
| **Twin** | | | | | | | | | |
| | Single | 19.7 | 0.542 | 15.0 | 0.6 | 12.7 | 0.127 | 6.2 | 0.951 |
| | Multiple | 16.5 | | 11.9 | | 21.5 | | 6.1 | |
| **Still breastfeeding** | | | | | | | | | |
| | No | 16.9 | <0.001 | 12.2 | <0.001 | 11.8 | 0.021 | 4.8 | <0.001 |
| | Yes | 21.8 | | 17.7 | | 14.2 | | 8.7 | |
| **Measles' vaccine** | | | | | | | | | |

*(Continued)*

**Table 3.** (Continued)

| Variables | | 2005 (n = 7,220) | | 2010 (n = 7,758) | | 2014 (n = 7,010) | | 2022 (n = 7,754) | |
|---|---|---|---|---|---|---|---|---|---|
| | | % Diarrhea | P value | % Diarrhea | P value | % Diarrhea | P value | Diarrhea | P value |
| | Not vaccinated | 21.4 | 0.02 | 18.2 | <0.001 | 16.3 | <0.001 | 4.9 | <0.001 |
| | Vaccinated | 18.7 | | 13.6 | | 11.5 | | 8.0 | |
| **Vitamin A supplement** | | | | | | | | | |
| | No | 18.5 | 0.004 | 16.0 | 0.132 | 12.1 | 0.301 | 5.9 | 0.261 |
| | Yes | 22.1 | | 14.4 | | 13.3 | | 6.7 | |
| **Household Characteristics** | | | | | | | | | |
| **Wealth index** | | | | | | | | | |
| | Poorest | 22.6 | <0.001 | 18.5 | <0.001 | 16.1 | 0.012 | 8.2 | <0.001 |
| | Poorer | 20.9 | | 15.9 | | 11.8 | | 8.4 | |
| | Middle | 19.8 | | 15.2 | | 10.5 | | 5.1 | |
| | Richer | 18.4 | | 12.1 | | 13.7 | | 5.1 | |
| | Richest | 14.3 | | 10.9 | | 11.4 | | 3.9 | |
| **Place of residence** | | | | | | | | | |
| | Urban | 16.6 | 0.027 | 10.6 | <0.001 | 12.7 | 0.926 | 4.5 | 0.002 |
| | Rural | 20.2 | | 15.8 | | 12.9 | | 7.2 | |
| **Region** | | | | | | | | | |
| | Phnom Penh | 19.0 | <0.001 | 11.9 | 0.017 | 17.7 | <0.001 | 2.5 | 0.006 |
| | Plain | 21.7 | | 15.6 | | 10.8 | | 6.8 | |
| | Tonle Sap | 19.1 | | 16.6 | | 13.7 | | 6.9 | |
| | Coastal | 10.4 | | 14.9 | | 9.7 | | 5.2 | |
| | Mountain | 20.7 | | 11.0 | | 14.7 | | 7.2 | |
| **Drinking water** | | | | | | | | | |
| | Improved | 20.2 | 0.354 | 14.7 | 0.648 | 11.5 | 0.022 | 5.6 | 0.055 |
| | Unimproved | 19.1 | | 15.3 | | 14.2 | | 7.0 | |
| **Toilet facility** | | | | | | | | | |
| | Improved | 14.6 | <0.001 | 12.0 | <0.001 | 12.1 | 0.205 | 6.1 | 0.7 |
| | Non-Improved | 21.1 | | 16.6 | | 13.7 | | 6.5 | |

Notes: Survey weights applied to obtain weighted percentages across survey years and combined data.

*Phnom Penh capital city; Plains: Kampong Cham, Tbong Khmum, Kandal, Prey Veng, Svay Rieng, and Takeo; Tonle Sap: Banteay Meanchey, Kampong Chhnang, Kampong Thom, Pursat, Siem Reap, Battambang, Pailin, and Otdar Meanchey; Coastal/sea: Kampot, Kep, Preah Sihanouk, and Koh Kong; Mountains: Kampong Speu, Kratie, Preah Vihear, Stung Treng, Mondul Kiri, and Ratanak Kiri.

children born to mothers who reported at least one barrier accessing to healthcare services were at increased risk (AOR = 1.20; 95% CI: 1.07–1.35) compared to those without barriers. Children aged 6–11 months (AOR = 2.05; 95% CI: 1.66–2.52), 12–23 months (AOR = 1.68; 95% CI: 1.39–2.04) also had higher odds of having diarrhea in the past two weeks.

## Discussion

We have described the temporal and geospatial prevalence of diarrhea among children in Cambodia between 2005 and 2022, noting a significant decrease from 19.7% in 2005 to 15.0% in 2010 to 12.9%% in 2014 and 6.2% in 2022. This reduction in diarrhea burden corresponds with global efforts to reduce under-five mortality, and likely contributed to Cambodia's achievement of Millennium Development Goal 4 (MDG-4) to reduce child mortality [6, 17–19]. This might be attributed to the efforts and initiative of the Royal Government of Cambodia (RGC), which has strengthened health facilities across the country, particularly in rural

**Table 4. Unadjusted and adjusted analyses of child diarrhea in Cambodia.**

| Variables | | 2005–2022 (n = 29,742) | | 2005–2022 (n = 29,742) | | 2005–2022 (n = 29,742) | |
|---|---|---|---|---|---|---|---|
| | | % Diarrhea | P value | OR | 95% CI | AOR | 95% CI |
| **Maternal Characteristic** | | | | | | | |
| **Age (years)** | | | | | | | |
| | <19 | 21.6 | <0.001 | Ref. | | Ref. | |
| | 19–24 | 17.1 | | 0.75 | (0.54–1.04) | 0.85 | (0.61–1.20) |
| | 25–34 | 12.2 | | 0.51*** | (0.37–0.70) | 0.68* | (0.48–0.96) |
| | 35–49 | 11.4 | | 0.47*** | (0.34–0.65) | 0.60** | (0.42–0.87) |
| **Educational** | | | | | | | |
| | No education | 16.2 | <0.001 | Ref. | | Ref. | |
| | Primary | 14.6 | | 0.89* | (0.79–0.99) | 0.99 | (0.88–1.11) |
| | Secondary | 10.1 | | 0.58*** | (0.51–0.67) | 0.84* | (0.72–0.98) |
| | Higher | 5.4 | | 0.29*** | (0.20–0.43) | 0.61* | (0.41–0.91) |
| **Employment** | | | | | | | |
| | Not working | 15.1 | <0.001 | Ref. | | Ref. | |
| | Working | 12.3 | | 0.79*** | (0.72–0.87) | 0.87** | (0.79–0.96) |
| **Smoking** | | | | | | | |
| | No-smoker | 13.1 | <0.001 | Ref. | | Ref. | |
| | Smoker | 21.9 | | 1.88*** | (1.49–2.36) | 1.61*** | (1.25–2.08) |
| **Media exposure** | | | | | | | |
| | No | 13.4 | 0.188 | Ref. | | | |
| | Yes | 11.5 | | 0.84 | (0.65–1.09) | | |
| **Healthcare Barriers** | | | | | | | |
| | No barrier | 9.7 | <0.001 | Ref. | | Ref. | |
| | 1 or More Barriers | 14.7 | | 1.61*** | (1.44–1.79) | 1.20** | (1.07–1.35) |
| **Child Characteristics** | | | | | | | |
| **Age(months)** | | | | | | | |
| | 0-5m | 12.6 | <0.001 | Ref. | | Ref. | |
| | 6-11m | 22.4 | | 2.00*** | (1.64–2.45) | 2.05*** | (1.66–2.52) |
| | 12-23m | 19.2 | | 1.64*** | (1.38–1.96) | 1.68*** | (1.39–2.04) |
| | 24-35m | 13.2 | | 1.06 | (0.89–1.26) | 1.04 | (0.85–1.26) |
| | 36-47m | 8.7 | | 0.65*** | (0.54–0.80) | 0.64*** | (0.52–0.79) |
| | 48-60m | 7.0 | | 0.52*** | (0.43–0.64) | 0.52*** | (0.42–0.66) |
| **Sex** | | | | | | | |
| | Boys | 14.3 | <0.001 | Ref. | | Ref. | |
| | Girls | 12.3 | | 0.84*** | (0.77–0.91) | 0.83*** | (0.76–0.91) |
| **Birth order** | | | | | | | |
| | 1 child | 13.5 | 0.002 | Ref. | | Ref. | |
| | 2–3 child | 12.6 | | 0.92 | (0.84–1.01) | 1.0 | (0.90–1.12) |
| | 4–5 child | 14.1 | | 1.05 | (0.92–1.21) | 1.02 | (0.86–1.20) |
| | ≥ 6 child | 16.2 | | 1.24** | (1.06–1.45) | 1.08 | (0.88–1.34) |
| **Twin** | | | | | | | |
| | Single | 13.3 | 0.649 | Ref. | | | |
| | Multiple | 14.6 | | 1.11 | (0.70–1.75) | | |
| **Still breastfeeding** | | | | | | | |
| | No | 10.7 | <0.001 | Ref. | | Ref. | |
| | Yes | 16.4 | | 1.64*** | (1.50–1.79) | 0.89* | (0.80–0.99) |
| **Measles' vaccine** | | | | | | | |

*(Continued)*

**Table 4.** (Continued)

| Variables | | 2005–2022 (n = 29,742) | | 2005–2022 (n = 29,742) | | 2005–2022 (n = 29,742) | |
|---|---|---|---|---|---|---|---|
| | | % Diarrhea | P value | OR | 95% CI | AOR | 95% CI |
| | Not vaccinated | 13.3 | 0.898 | Ref. | | | |
| | Vaccinated | 13.3 | | 1.01 | (0.92–1.10) | | |
| **Vitamin A supplement** | | | | | | | |
| | No | 12.8 | 0.055 | Ref. | | Ref. | |
| | Yes | 13.8 | | 1.09 | (1.00–1.20) | 1.1 | (1.00–1.22) |
| **Household Characteristics** | | | | | | | |
| **Wealth index** | | | | | | | |
| | Poorest | 16.7 | <0.001 | Ref. | | Ref. | |
| | Poorer | 14.6 | | 0.85* | (0.75–0.97) | 0.9 | (0.79–1.03) |
| | Middle | 12.5 | | 0.71*** | (0.62–0.82) | 0.77*** | (0.67–0.89) |
| | Richer | 11.7 | | 0.66*** | (0.57–0.77) | 0.79** | (0.66–0.93) |
| | Richest | 9.8 | | 0.54*** | (0.47–0.64) | 0.69*** | (0.55–0.86) |
| **Place of residence** | | | | | | | |
| | Urban | 9.0 | <0.001 | Ref. | | Ref. | |
| | Rural | 14.5 | | 1.71*** | (1.49–1.97) | 1.13 | (0.98–1.30) |
| **Region** | | | | | | | |
| | Phnom Penh | 10.9 | 0.002 | Ref. | | Ref. | |
| | Plain | 13.9 | | 1.33* | (1.06–1.67) | 0.79* | (0.62–0.99) |
| | Tonle Sap | 14.0 | | 1.34** | (1.07–1.67) | 0.75* | (0.60–0.94) |
| | Coastal | 10.1 | | 0.93 | (0.72–1.19) | 0.53*** | (0.41–0.69) |
| | Mountain | 13.2 | | 1.25 | (0.99–1.57) | 0.67** | (0.53–0.85) |
| **Drinking water** | | | | | | | |
| | Improved | 12.7 | 0.018 | Ref. | | Ref. | |
| | Unimproved | 14.1 | | 1.13* | (1.02–1.24) | 1.01 | (0.91–1.12) |
| **Toilet facility** | | | | | | | |
| | Improved | 9.6 | <0.001 | Ref. | | Ref. | |
| | Non-Improved | 16.9 | | 1.93*** | (1.74–2.14) | 1.07 | (0.93–1.23) |
| **Survey years** | | | | | | | |
| | 2005 | 19.7 | <0.001 | Ref. | | Ref. | |
| | 2010 | 15.0 | | 0.72*** | (0.63–0.82) | 0.74*** | (0.64–0.85) |
| | 2014 | 12.9 | | 0.60*** | (0.53–0.69) | 0.62*** | (0.54–0.72) |
| | 2022 | 6.2 | | 0.27*** | (0.23–0.31) | 0.32*** | (0.26–0.38) |

* $p < 0.05$

** $p < 0.01$

*** $p < 0.001$

areas, improved the medical infrastructure, provided essential medical equipment and supplies, and improved safe drinking water and sanitation in rural area [6]. Recently, the RGC has prioritized child health and nutrition, closely monitored by the Prime Minister to ensure policy implementation and achieve Sustainable Development Goals 3 (SDGs-3) [20–22]. Cambodia has set ambitious goals within SDG-3, specifically aiming to reduce neonatal mortality to less than 12 per 1,000 live births and under-five mortality to less than 25 per 1,000 live births by 2030. These goals require continued focus on preventable childhood deaths, including diarrhea.

Our study found several factors associated with increased odds of having diarrhea among children aged 0–59 months in Cambodia. These include children born to mothers who smoke, mothers with limited access to healthcare services, and children aged 6–23 months. However, children were less likely to have diarrhea if their mothers were 25 or older, had at least a secondary education, were employed, or if the child was girl, aged 36–60 months, lived in the richest wealth quintile, resided in a coastal region, or were surveyed in later years.

Children of mothers aged 25–34 years and 35–49 years have shown less occurrence of childhood diarrhea than young mothers. This is consistent with previous studies in Cambodia and India [10, 17]. Likely, lower knowledge and experience of childcare and feeding among younger mothers can be a plausible causal explanation [10, 18–20]. The odds of having diarrhea are lower among the children of working mothers, and similar findings have been identified in some recent studies in Cambodia, India, and Senegal [10, 17, 21]. Several studies document that the mother's employment status may have an education, which will improve a child's quality of living environment standards as well as improve hygienic practice, safe drinking water, and sanitation in the home during feeding and childcare [10, 15, 22, 23]. Children of mothers completed secondary or higher education were found to be less likely than children whose mothers had no education to experience diarrhea. This is in line with findings from Cambodia [15, 22]. This finding might be explained by the fact that highly educated mothers might have access to books and more education education programs that help to better knowledge and experience of childcare and feeding children [15, 22].

Children from wealthier households were less likely to have diarrhea compared to those from poorer households, consistent with studies in Bangladesh, India, and Cambodia [15, 17, 22, 24]. This likely reflects increased access to nutrition and healthcare for children in wealthier households [15, 17, 22]. Higher income allows families to provide a more diverse diet, improved quality of care, and access to health services [25]. In this study, the age of the children significantly affected their risk of diarrhea. Children aged 36–47 months (AOR = 0.64; 95% CI: 0.52–0.79) and 48–60 months (AOR = 0.52; 95% CI: 0.42–0.66) had a lower odds ratio of having diarrhea compared to those aged 0–5 months. This finding aligns with similar studies [17, 22]. However, children aged 6–11 months (AOR = 2.05; 95% CI: 1.66–2.52) and 12–23 months (AOR = 1.68; 95% CI: 1.39–2.04) were more likely to have diarrhea compared to infants under five months. This aligns with research conducted in Northwest Ethiopia, India, and Cambodia [15, 17, 22, 23] and might be due to the developing immune system throughout childhood [22]. Girls were less likely to have diarrhea than boys, which may reflect a natural predisposition of males to develop diarrhea more frequently than girls [9].

Children of smoking mothers were 1.61 times more likely to have diarrhea compared to children of non-smoking mothers. A narrative review focused on smoking increases the risk of infectious diseases, which found that investigators found that smoking was the leading risk factor in a reduced immune response [24]. Children of mothers having a problem accessing healthcare services were 1.20 times more likely to have diarrhea compared to children of mothers not having a problem. This aligns with findings from 31 countries in sub-Saharan Africa [25]. For women, access to medical care is one buffer against poor health. However, disadvantaged women experience limited access to care and are more likely to have an unplanned pregnancy, partly reflecting inadequate preconception care and family planning. They are also more likely to start prenatal care later than advantaged peers, deliver in hospitals with lower quality-of-care indicators, and are less likely to know about childcare [25]. Lastly, children living in rural areas had AOR = 1.13 times more likely to have diarrhea than in urban areas. This result was in line with the study conducted in India and Cambodia [10, 17]. This could be due to limited access to healthcare and sanitation facilities in rural areas [15, 22, 23]. Thus, compared to children living in households with improved toilet facilities, children from households

with unimproved toilets were more likely to experience diarrhea. Improved sanitation facilities were found in a supporting multicounty WASH-intervention study to reduce the risk of children from fever by 13% and cough by 10% [26, 27].

## Strengths and limitations

This study has several strengths; it used combined data from Cambodia Demographic and Health surveys from 2005, 2010, 2014, and 2022. This enabled us to describe the temporal and geographical trends in diarrhea among children aged 0–59 months using nationally representative data [15, 16]. Likewise, we were able to examine factors associated with diarrhea among children throughout Cambodia. We acknowledge that the results from this study may reflect the present prevalence of diarrhea among children in the country by comparing the associated factors with diarrhea among children to another DHS survey. It is hoped that the evidence will help the Cambodian Ministry of Health's Child Health program, as well as health promotion and social determinants, to plan interventions to reduce child morbidity and mortality due to diarrhea. Despite the strengths, the CDHS collected data as a cross-sectional study, so examining causality in these relationships was impossible. Also, the outcome variable was collected from self-reporting from mothers or caregivers to define the presence of diarrhea among eligible children in the past two weeks before data collection, making our analysis prone to recall bias. However, this recall bias appears non-differential since the DHS survey uses the standard questionnaire recommended by UNICEF/WHO [28]. As a secondary data analysis, several known etiologies of diarrhea, such as parasites, viruses, bacteria, and common factors for diarrhea. These included children's malnutrition, history of illness, antibiotic prescription, type of roofing material, household types of cooking, season effect, and protective vaccination against diarrhea, which were not included in this study due to data limitations.

## Conclusion

Our findings suggest that the relatively high prevalence of diarrhea among children aged 0–59 months in Cambodia declined slightly over the 20 years from 19.7% in 2005 to 6.2% in 2022. Diarrhea among children continues to be a significant public health concern in Cambodia, particularly in specific regions and among vulnerable children and those living in remote areas. The mountainous regions, such as Mondul Kiri and Ratanak Kiri, and the plain region consistently had higher rates of diarrhea across all survey years. We also found factors independently associated with increased diarrhea risk include child age (mainly aged 6–23 months), maternal smoking, and barriers to healthcare access, with these factors remaining consistent over time. On the other hand, children of mothers with secondary education or higher, mothers in employment, and those from wealthier households were less likely to suffer from diarrhea. Public health practitioners and child health policymakers should focus on these vulnerable groups—mainly children aged 6–23 months living in remote, mountainous provinces and those in low-income households with limited educated mothers and reported having buried healthcare accessible interventions that improve access to healthcare, enhance maternal education, and promote better hygiene and sanitation practices are critical important role for reducing the burden of diarrhea among children in Cambodia.

## Supporting information

**S1 Table. Results of checking multicollinearity using variance inflation factor.**
(DOCX)

**S1 Dataset.**

(DTA)

## Acknowledgments

The authors would like to thank DHS-ICF, who approved the data used for this paper.

## Author Contributions

**Conceptualization:** Samnang Um.

**Data curation:** Samnang Um.

**Formal analysis:** Samnang Um.

**Investigation:** Samnang Um.

**Methodology:** Samnang Um.

**Project administration:** Samnang Um.

**Resources:** Samnang Um.

**Software:** Samnang Um.

**Supervision:** Samnang Um.

**Validation:** Samnang Um.

**Visualization:** Samnang Um.

**Writing – original draft:** Samnang Um, Channnarong Phan, Sok Sakha, Leng Dany.

**Writing – review & editing:** Samnang Um.

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
