## [Decision Letter · Decision Letter 0]

6 Oct 2024

PONE-D-24-17960Child diarrhea in Cambodia: A descriptive analysis of temporal and geospatial trends and logistic regression-based examination of factors associated with diarrhea in children under five yearsPLOS ONE

Dear Dr. Um,

Thank you for submitting your manuscript to PLOS ONE. After careful consideration, we feel that it has merit but does not fully meet PLOS ONE’s publication criteria as it currently stands. Therefore, we invite you to submit a revised version of the manuscript that addresses the points raised during the review process.In the abstract, the authors write “…used multivariate logistic regression…”. I think they used multivariable logistic regression. There is a huge difference between “multivariable” and “multivariate”. Check it carefully.In logistic regression, the authors use pooled data. Justify the use of combined data.I think, in this circumstance, spatiotemporal modeling is more effective than applying logistic regression in combined data.In Table 2, just keep the “% Diarrhea” column. No need to add “% Not Diarrhea”. It seems redundant because “% diarrhea + % Not diarrhea=100”.

We look forward to receiving your revised manuscript.

Kind regards,

Md. Moyazzem Hossain

Academic Editor

PLOS ONE

Journal Requirements:

3. We note that [Figure 4] in your submission contain [map/satellite] images which may be copyrighted. All PLOS content is published under the Creative Commons Attribution License (CC BY 4.0), which means that the manuscript, images, and Supporting Information files will be freely available online, and any third party is permitted to access, download, copy, distribute, and use these materials in any way, even commercially, with proper attribution. For these reasons, we cannot publish previously copyrighted maps or satellite images created using proprietary data, such as Google software (Google Maps, Street View, and Earth). For more information, see our copyright guidelines: http://journals.plos.org/plosone/s/licenses-and-copyright.

a. You may seek permission from the original copyright holder of Figure 4 to publish the content specifically under the CC BY 4.0 license. 

Reviewers' comments:

Reviewer's Responses to Questions

**Comments to the Author**

1. Is the manuscript technically sound, and do the data support the conclusions?

Reviewer #1: Partly

Reviewer #2: Yes

2. Has the statistical analysis been performed appropriately and rigorously? 

Reviewer #1: Yes

Reviewer #2: Yes

3. Have the authors made all data underlying the findings in their manuscript fully available?

Reviewer #1: Yes

Reviewer #2: Yes

4. Is the manuscript presented in an intelligible fashion and written in standard English?

Reviewer #1: No

Reviewer #2: Yes

5. Review Comments to the Author

Reviewer #1: 1- Mismatch between title and results: Authors put the title “Child diarrhea in Cambodia: A descriptive analysis of temporal and geospatial trends and logistic regression-based examination of factors associated with diarrhea in children under five years”, but the results partially match with the title. The results focus primarily on temporal trends in diarrhea prevalence, but did not examine the geographic variations of diarrhea as state in the title. Authors should revise the title accordingly.

2- English writing: Authors should thoroughly review their English throughout the manuscript. For instance, from line 17 to 20 “The highest prevalence …… 2014 and 2022”, there is no verb.

3- Line 20-27: The presence of both odds ratios (ORs) less than 1 and greater than 1 indicates a relationship between the studied factors and childhood diarrhea. To avoid confusion, a more nuanced interpretation is necessary. Simply stating factors were 'independently associated' with diarrhea, like 'mothers aged 25-34 years (adjusted odds ratio (AOR) = 0.68; 95% CI: 0.48–0.96)', doesn't clarify tell the audiences whether this association is protective (OR < 1) or increases the risk (OR > 1) of diarrhea. Authors should provide a clearer explanation of the direction of the associations.

4- Line 27-29: Recommendation is too broad. Authors should provide more precise recommendation based on the findings.

5- Line 51-58: The manuscript focuses primarily on diarrhea. However, paragraph from line 51-58 are about under-five mortality which is a broader topic. It would be beneficial if the authors could address the specific contribution of diarrhea to overall under-five mortality within the paragraph. If data on this attribution is unavailable, it's best to omit this section entirely.

6- Line 134: Please omit “,” after “Plains included”

7- Line 133-138: it would be easy to understand if authors puts the 5 geographical regions in a table.

8- Line 147: “…. variations in the prevalence of anemia over time……” What does it mean?

9- Table 1: “Had diarrhea in the past two weeks”. I assume that this refers to children. If so, please specify. To be easily understand the table, authors may capitalize main titles: “Had diarrhea in the past two weeks”, “Maternal Characteristics”, “Child Characteristics”, and “Household Characteristics”

10- Line 286-288: Please rewrite the sentence

11- From Discussion part afterward, it is difficult to provide feedback to a specific part as there are no line numbers. Sometime, authors use the word “male and female” which are not appropriate for this age group.

12- In conclusion part: Authors should highlight factors associated with diarrhea across the survey years and/or geographical regions to reflex the title and research finding, rather than just mentioning x, y, … factors were associated with diarrhea.

Reviewer #2: The Authors have addressed a very important topic of childhood Diarrhoea Cambodia, Diarrhoea being a major cause of Morbidity and mortality among U5s in LMICs

This is a well-written paper in the Abstract the authors provided the prevalence of Diarrhoea but didn’t indicate the total sample size of children analyzed. This should be done. The second sentence of the abstract is too long. It should be revised accordingly.

The results were fairly well presented but tables 1,2 and 3 are very busy and long some covering 3 pages! Is here away these can be reduced e.g. some of the information written in text form. The table, graph and chats in the appendix are not numbered not titled. This should be done. The authors discussed their results well and made appropriate conclusions’ limitations should be revised to make them clear and concise. Did their study have any strength. They only talked about the study limitations.

6. PLOS authors have the option to publish the peer review history of their article (what does this mean?). If published, this will include your full peer review and any attached files.

Reviewer #1: No

Reviewer #2: **Yes: **Prof. Edison Arwanire Mworozi

---

## [Author Response · Author response to Decision Letter 0]

4 Nov 2024

We thank the editor and all reviewers for the opportunity to revise and resubmit our manuscript. All of the reviewers provided us with thoughtful and thorough evaluations of our paper. We believe that the revisions we have made based on these comments have allowed us to develop a much-improved manuscript. We feel that we have addressed all of the reviewer’s comments. However, if we have missed something, we are more than willing to make the necessary changes going forward. Our responses to the specific points raised by each reviewer are outlined in the summary of changes outlined herein. In addition, we have uploaded a minimal dataset with a file extension in STATA format and further revised the source of the Cambodia Shapefiles, which were obtained from the the United Nations Office for the Coordination of Humanitarian Affairs (OCHA) and is accessible at https://data.humdata.org/dataset/cod-ab-khm.

Best regards, 

Samnang

---

## [Decision Letter · Decision Letter 1]

8 Dec 2024

Child diarrhea in Cambodia: A descriptive analysis of temporal and geospatial trends and logistic regression-based examination of factors associated with diarrhea in children under five years

PONE-D-24-17960R1

Dear Dr. Um,

We’re pleased to inform you that your manuscript has been judged scientifically suitable for publication and will be formally accepted for publication once it meets all outstanding technical requirements.

Kind regards,

Md. Moyazzem Hossain, PhD

Academic Editor

PLOS ONE

Additional Editor Comments (optional):

Reviewers' comments:

Reviewer's Responses to Questions

**Comments to the Author**

1. If the authors have adequately addressed your comments raised in a previous round of review and you feel that this manuscript is now acceptable for publication, you may indicate that here to bypass the “Comments to the Author” section, enter your conflict of interest statement in the “Confidential to Editor” section, and submit your "Accept" recommendation.

Reviewer #1: All comments have been addressed

Reviewer #2: All comments have been addressed

2. Is the manuscript technically sound, and do the data support the conclusions?

Reviewer #1: Yes

Reviewer #2: Yes

3. Has the statistical analysis been performed appropriately and rigorously? 

Reviewer #1: Yes

Reviewer #2: Yes

4. Have the authors made all data underlying the findings in their manuscript fully available?

Reviewer #1: Yes

Reviewer #2: Yes

5. Is the manuscript presented in an intelligible fashion and written in standard English?

Reviewer #1: Yes

Reviewer #2: Yes

6. Review Comments to the Author

Reviewer #1: Author has revised the manuscript as per comments and suggestion. Overall, the manuscript is scientifically sound.

Reviewer #2: his paper presents a comprehensive analysis of diarrhea in children under five years in Cambodia from 2005 to 2022. The authors utilize data from the Cambodia Demographic and Health Surveys to examine temporal and geospatial trends in diarrhea prevalence.

The study reveals a significant decline in diarrhea incidence over the study period, highlighting the impact of public health interventions. However, disparities persist across regions, with some areas experiencing higher prevalence rates.

The authors also explore factors associated with diarrhea, including socioeconomic and environmental factors. They emphasize the importance of continued efforts to improve sanitation, access to clean water, and hygiene practices to further reduce the burden of diarrheal disease in Cambodia.

This research provides valuable insights into the epidemiology of diarrhea in Cambodia and underscores the need for targeted interventions to address regional disparities and promote child health.

7. PLOS authors have the option to publish the peer review history of their article (what does this mean?). If published, this will include your full peer review and any attached files.

Reviewer #1: **Yes: **Yom An

Reviewer #2: No

---

## [Editor Report · Acceptance letter]

10 Jan 2025

PONE-D-24-17960R1 

PLOS ONE

Dear Dr. Um, 

I'm pleased to inform you that your manuscript has been deemed suitable for publication in PLOS ONE. Congratulations! Your manuscript is now being handed over to our production team.

Kind regards, 

on behalf of

Professor Md. Moyazzem Hossain 

Academic Editor

PLOS ONE